# Adult Onset of Type 2 Familial Hemophagocytic Lymphohistiocytosis After SARS-CoV-2 Vaccination with an Unusual Neurological Onset: The Great Mimic

**DOI:** 10.3390/diagnostics15233000

**Published:** 2025-11-26

**Authors:** Flaminia Bellisario, Assunta Bianco, Francesco D’Alo’, Chiara Passarelli, Rosellina Russo, Massimiliano Mirabella, Simona Sica, Stefan Hohaus

**Affiliations:** 1UOSD Malattie Linfoproliferative Extramidollari, Dipartimento di Scienze di Laboratorio ed Ematologiche , Fondazione Policlinico Universitario A. Gemelli IRCCS, 00168 Roma, Italy; 2Multiple Sclerosis Center, Fondazione Policlinico Universitario A. Gemelli IRCCS, 00168 Roma, Italy; 3Unità di Ricerca di Citogenomica Traslazionale, UOC Laboratorio di Genetica Medica, IRCCS Ospedale Pediatrico del Bambino Gesù, 00165 Roma, Italy; 4UOSD Neuroradiologia Diagnostica, Dipartimento di Diagnostica per Immagini e Radioterapia Oncologica, Fondazione Policlinico Universitario A. Gemelli IRCCS, 00168 Roma, Italy; 5Centro di Ricerca per la Sclerosi Multipla “Anna Paola Batocchi”, Università Cattolica del Sacro Cuore, 00168 Roma, Italy; 6Dipartimento di Scienze di Laboratorio ed Ematologiche, UOC Ematologia e Trapianto di Cellule Staminali Emopoietiche, Fondazione Policlinico Universitario A. Gemelli IRCCS, 00168 Roma, Italy

**Keywords:** hemophagocytic lymphohistiocytosis, ADEM, SARS-CoV-2 vaccination, PRF1 mutation, HSCT

## Abstract

**Background and Clinical Significance:** This case report describes a 46-year-old male with no prior comorbidities who developed progressive neurological symptoms—ataxia and diplopia—shortly after the second Comirnaty (Pfizer-BioNTech) COVID-19 vaccine dose. The aim is to highlight the diagnostic challenges of central nervous system-dominant hemophagocytic lymphohistiocytosis (HLH) and its overlap with neuroinflammatory disorders. **Case Presentation:** Initial MRI showed demyelinating lesions in the brain and spinal cord, suggesting acute disseminated encephalomyelitis (ADEM). The patient had only transient improvement with corticosteroids and then multiple relapses with expanding CNS lesions despite cyclophosphamide, plasmapheresis, and rituximab. After 27 months, systemic features appeared, including fever, cytopenias, elevated inflammatory markers, and splenomegaly. Bone marrow analysis revealed hemophagocytosis, fulfilling HLH-2004 criteria, with an H-score of 200 supporting secondary HLH. Given consanguinity and persistent immune activation, next-generation sequencing identified two homozygous PRF1 variants—one pathogenic (p.Arg232His) and one of uncertain significance (p.Ala91Val)—consistent with autosomal recessive familial type 2 HLH. The patient underwent matched unrelated donor hematopoietic stem cell transplantation (HSCT) 11 months after HLH diagnosis, achieving initial stabilization, but ultimately died from infectious complications in March 2025 without evidence of HLH relapse. **Conclusions:** This case illustrates an atypical adult-onset presentation of familial HLH manifesting primarily with recurrent neuroinflammatory symptoms that initially mimicked ADEM. The diagnostic delay reflects the challenge of recognizing CNS-dominant HLH, especially in adults and in the absence of early systemic features. The identification of biallelic PRF1 variants confirmed an underlying genetic predisposition. This is the first reported case of adult-onset familial HLH presenting predominantly with neurological symptoms following COVID-19 vaccination. The case emphasizes the need to consider genetic forms of HLH in relapsing neuroinflammatory disorders and raises the hypothesis that vaccination may unmask subclinical immune dysregulation in genetically susceptible individuals

## 1. Introduction

Hemophagocytic lymphohistiocytosis (HLH) is a rare and life-threatening syndrome characterized by uncontrolled immune activation, excessive cytokine release, and macrophage hyperactivation. The condition arises from a failure of cytotoxic T lymphocytes and natural killer (NK) cells to regulate the immune response following a triggering event. HLH is classified into primary (genetic) and secondary (acquired) forms. Primary HLH is caused by pathogenic variants in one of nine genes involved in the assembly, trafficking, or exocytosis of cytolytic granules in CD8^+^ T cells and NK cells [1]. Approximately 70% of cases present within the first year of life, whereas adult-onset forms are uncommon.

A hallmark of the disease is the hyperactivation of cytotoxic T lymphocytes, which induces a cascade of systemic hyperinflammation and macrophage activation. Of particular importance is interferon-gamma (IFN-γ), which acts as a key mediator between lymphocytes and macrophages. Persistent antigen presentation and continuous IFN-γ stimulation sustain an exaggerated cytokine cascade, resulting in overproduction of multiple proinflammatory mediators, including IFN-γ, TNF-α, IL-1, IL-4, IL-6, IL-8, IL-10, and IL-18 [2,3]. Concomitantly, impaired cytolytic function of NK and CD8^+^ T cells, caused by defects in granule formation, trafficking and exocytosis, prevents efficient lysis of infected or antigen-presenting cells. This leads to prolonged cell–cell interactions and amplification of the proinflammatory cytokine storm.

The diagnosis of HLH is based on diagnostic scoring systems. The HLH-2004 criteria are generally applied in pediatric cases, whereas the H-score is typically employed to assess secondary forms.

The main clinical manifestations of HLH include fever, hepatosplenomegaly, lymphadenopathy, cytopenia, hyperferritinemia, hypertriglyceridemia, hypofibrinogenemia and multiorgan dysfunction, which may also lead to neurological symptoms [4].

The therapeutic approaches most employed in adults are adapted from pediatric treatment protocols [5]. In approximately 20–30% of adult cases, HLH is either refractory to first-line therapy or relapses after an initial remission. There remains no standardized therapeutic approach for patients with relapsed or refractory primary HLH (pHLH), and long-term cure is typically achieved only through allogeneic hematopoietic stem cell transplantation (allo-HSCT). However, treatment-related mortality remains high, with up to 25% of patients unable to reach transplantation [6].

Given the pivotal role of excessive immune activation and cytokine dysregulation in the pathogenesis of HLH, several molecular and immunologic targets have been proposed in recent years. These include interferon-gamma (IFN-γ), the Janus kinase–signal transducer and activator of transcription (JAK–STAT) pathway, interleukins IL-6, IL-1, and IL-18, tumor necrosis factor-alpha (TNF-α), and cell-surface molecules such as CD52, CD20, and programmed cell death protein 1 (PD-1) [7].

## 2. Case Report

The clinical case involves a 46-year-old male patient with no significant comorbidities, born to first-cousin parents and with a family history notable for hematological disease, as his father had died of acute myeloid leukemia. In July 2021, three weeks after receiving the second dose of the SARS-CoV-2 vaccine (Pfizer Comirnaty BNT162b2) on 28 June, he developed progressive balance impairment with gait ataxia, leading to unsteadiness while walking and frequent falls. By September 2021, he reported transient episodes of diplopia, which subsequently became persistent. Neurological examination at that time revealed truncal ataxia, impaired tandem walking, mild limb incoordination, and a mild right sixth cranial nerve (abducens) deficit.

Magnetic Resonance Imaging (MRI) revealed multiple white matter lesions, hyperintense on FLAIR sequences and contrast-enhancing on T1-weighted images, with an asymmetric and periventricular distribution in the brain and spinal cord (Figure 1).

The patient was treated with intravenous methylprednisolone (1 g daily for five days), followed by a tapering course of oral prednisone (50 mg daily). leading to significant clinical improvement. He remained clinically and radiologically stable until July 2022, when he developed acute gait disturbances and ambulation difficulty.

Upon admission to the Neurology Department, a new MRI showed a “mass-like” lesion in the left paratrigonal region with hemorrhagic components, another at the C6 spinal level, and multiple contrast-enhancing lesions were identified in the cerebellum, supratentorial white matter, and spinal cord (Figure 2).

Testing for anti-MOG and anti-AQP4 antibodies was negative. Cerebrospinal fluid (CSF) analysis revealed mild hyperproteinorrachia (protein level: 45 mg/dL) without pleocytosis. Microbiological and oligoclonal band analysis were negative. Serological autoimmune and thrombophilia workups were unremarkable. Visual evoked potentials were within normal limits.

Due to rapidly increasing disability and aggressive disease progression, the patient received intravenous methylprednisolone (1 g daily for five days) and cyclophosphamide (total of 5 g administered from July 2022 to January 2023), resulting in clinical and radiological stabilization.

However, in February 2023, cyclophosphamide was discontinued due to liver toxicity, and mycophenolate mofetil (2 g daily) was initiated. By May 2023, worsening neurological symptoms consisting of gait disturbances and a left-sided sensorimotor hemisyndrome prompted an additional MRI, which demonstrated extensive new FLAIR hyperintense and contrast-enhancing lesions in the deep white matter and spinal cord (Figure 3). Limited response to intravenous methylprednisolone (1 g daily for five days) led to five sessions of plasmapheresis, which significantly improved clinical symptoms. Treatment with rituximab (1 g on days 1 and 15) was started in July 2023.

In October 2023, the patient was admitted with asthenia, worsening gait, fever, persistent constipation and pharyngodynia. Notably, for the first time, he exhibited thrombocytopenia (platelets at 7000/mm^3^), lymphocytosis, and elevated levels of creatinine and transaminases. Laboratory findings included high levels of ferritin (2421 ng/mL), CRP (26.6 mg/L), and procalcitonin (1.76 ng/mL). Bone marrow aspirate revealed hemophagocytosis. Abdominal ultrasound showed mild splenomegaly (13 cm). Intravenous corticosteroids were promptly initiated with clinical and laboratory improvement. In March 2024, the patient was readmitted for pneumonia. For the first time, significant splenomegaly (20 cm) was detected. Despite infection resolution, follow-up tests revealed hypertransaminasemia (GOT 1039 UI/L, GPT 1409 UI/L, GGT 275 UI/L), hyperferritinemia (3100 ng/mL), increased CRP (637 mg/L), negative procalcitonin and elevated LDH levels. The complete blood count showed hemoglobin (Hb) of 12.7 g/dL, platelets at 76,000/mm^3^, and white blood cell (WBC) count of 2180/mm^3^.

Given these findings, the patient was admitted to the Hematology ward in April 2024. The patient was in fair general condition, afebrile, with known ambulatory deficits.

A total-body CT scan confirmed the persistence of pulmonary lesions, a spleen diameter of 15.5 cm, and a slightly enlarged liver without focal lesions. Therapy was initiated with dexamethasone (8 mg BID) and intravenous acetylcysteine (300 mg three times a day). A repeat bone marrow showed hemophagocytosis. Serologic tests for HIV, HAV, HBV, and HCV were negative. EBV-PCR test was performed at disease onset and monitored throughout the hematologic management, consistently yielding negative results up to February 2025. Laboratory tests showed ferritin at 902 ng/mL, triglycerides at 200 mg/dL, fibrinogen at 235 mg/dL, GOT at 86 UI/L, GPT at 1390 U/L, Hb at 7.5 g/dL, platelets at 46,000/mm^3^, WBC at 2210/mm^3^, and sCD25 > 7500 UI/L. During hospitalization, bronchoalveolar lavage was positive for HSV1 and weakly positive for Aspergillus, with a galactomannan index of 0.7. Consequently, antiviral therapy with acyclovir and antifungal therapy with isavuconazole were initiated. Due to persistent transaminase elevation, a liver biopsy was performed, revealing marked activation of Kupffer cells with erythrophagocytosis. Autoimmune panel tests, including anti-LKM, AMA, and ASMA antibodies, were negative. Repeated Quantiferon testing was also negative. Detailed laboratory analyses are provided in Appendix A.

Using clinical and laboratory data, we calculated the H-score, which had been developed by Fardet et al. [8] to help diagnose secondary hemophagocytic lymphohistiocytosis (sHLH). H-score was 200, corresponding to an 80–88% probability of sHLH. Given the uncertain trigger of immune activation and a history of parental consanguinity, next-generation sequencing (NGS) was performed to assess for mutations associated with primary HLH. Genetic analysis identified two homozygous variants in the PRF1 gene: c.272C>T (p.Ala91Val) and c.695G>A (p.Arg232His). The missense variant c.272C>T (p.Ala91Val) has a global allele frequency of 0.02916 (gnomAD), is the most prevalent variant found in this gene, and, even if with a controversial role, is still classified as a variant of uncertain significance (VUS, class 3) [9]. The c.695G>A (p.Arg232His) variant has an allelic frequency of 0.00007099 in the global population (gnomAD) and is classified as a pathogenic variant (class 5) according to ACMG guidelines. All other HLH-related genes analyzed (UNC13D, STX11, STXBP2, RAB27A, LYST, SH2D1A and XIAP) were negative for pathogenic or likely pathogenic variants.

Based on these findings, a diagnosis of type 2 familial HLH with autosomal recessive inheritance was established. Given the genetic findings, the patient was referred for an allogeneic hematopoietic stem cell transplant (HSCT).

While awaiting transplantation, the patient continued oral dexamethasone therapy with gradual tapering. Despite this approach, iatrogenic adrenal insufficiency developed and remained persistent.

A matched unrelated donor (MUD) was identified, and the patient underwent HSCT in November 2024. A follow-up MRI performed one month after the procedure showed near-complete resolution of spinal cord signal alterations and a significant reduction in enhancing brain lesions.

During hospitalization for HSCT, no severe complications occurred, and engraftment was achieved promptly. The patient was discharged while still experiencing walking difficulties due to prolonged immobilization.

In February 2025, the patient overcame a febrile episode related to SARS-CoV-2 infection with supportive therapy. Subsequently, he was rehospitalized due to fever and respiratory failure.

During the final hospitalization in March, follow-up MRI revealed early radiological signs of disease reactivation, without clear laboratory evidence of relapse (Figure 4).

The patient eventually passed away at the end of March 2025 due to post-transplant respiratory infectious complications (Figure 5).

**Figure 5 diagnostics-15-03000-f005:**
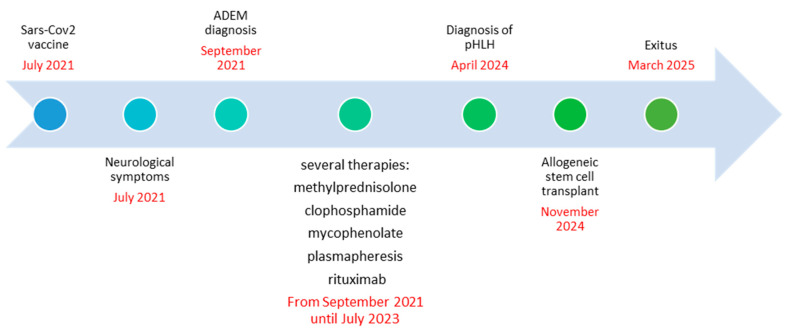
Timeline summarizing the patient’s disease progression and treatment course.

## 3. Discussion

The case report describes an adult-onset form of familial hemophagocytic lymphohistiocytosis (HLH) presenting initially with neurological symptoms following the second dose of the Comirnaty vaccine. The initial diagnosis was ADEM. Notably, a similar case has been reported of a 20-year-old patient, initially diagnosed with ADEM, later confirmed to have HLH due to a homozygous PRF1 gene mutation [10]. This patient presented with bilateral vision loss with MRI findings of demyelinating lesions in the cerebral white matter, particularly in the occipital lobes, corpus callosum, cerebellar hemispheres, and dorsal pons, with nodular and linear contrast enhancement. Despite initial treatment with methylprednisolone and plasmapheresis, the patient’s condition deteriorated, leading to the identification of HLH through genetic testing. Meeting the HLH-2004 criteria, the patient was subsequently referred for allogeneic transplant [9].

In the literature, seven additional cases of adult-onset familial HLH with neurological presentations have been reported. A retrospective study on adult-onset HLH, encompassing both primary and secondary forms, found CNS involvement in 10% (29 out of 289 patients) [11]. In pediatric HLH, CNS involvement rates range from 37% to 73% [12], with hemorrhagic lesions also reported. However, CNS-HLH incidence varies significantly, primarily due to reliance on retrospective data.

PRF1 dysfunction leads to excessive cytokine release and inability to switch off the immune response, resulting in damage to the blood–brain barrier. This increases permeability, permitting leukocytes, macrophages, and histiocytes to infiltrate the CNS and directly trigger inflammatory cascades affecting the myelin. Typically, the cerebrospinal fluid (CSF) is sterile with variable pleocytosis (10–47%) and protein elevation (11–41%) [13,14,15]. PRF1 mutations account for 20–40% of childhood primary HLH cases; their incidence in adults remains less well-characterized [16].

Genetic studies on adult-onset familial HLH reveal considerable variability. According to a review by Southam et al. [11], most cases involved mutations in the PRF1 gene (4 out of 8), typically in compound heterozygosity or double heterozygosity. PRF1 mutations manifest in various forms, including missense mutations, frameshift mutations, nonsense mutations, and in-frame insertions/deletions, with the first three being the most prevalent. Over 70 different mutations have been identified across the two coding exons of PRF1, with p.W374X being the most common. Geographic and racial variations in PRF1 mutations have also been documented [16,17,18].

In the study by Cetica et al., based on a large Italian registry investigating genetic predisposition to HH, the genes associated with familial forms of the disease were retrospectively analyzed in a cohort of 500 individuals, including 44 adult patients [19]. Consistent with previous literature, PRF1 was the most frequently affected gene, followed by UNC13D. The registry revealed a high prevalence of biallelic mutations (34%) but also underscored that genetic predisposition alone is insufficient for HLH development. Even in primary forms, an external trigger is required to initiate the hyperinflammatory response.

A study by Wang et al. [20] found that among 252 adult patients with HLH without a known family history of HLH, 7.1% (18 cases) exhibited genetic alterations linked to primary HLH. PRF1 mutations were present in 50% of these cases in both monoallelic and biallelic forms. These patients demonstrated reduced NK cell activity, yet prognosis did not differ based on mutation type or number. In patients with mutations associated with primary HLH, an allogeneic transplant was pursued.

Similarly, another study found that UNC13D mutations are more frequently implicated in the onset of HLH compared to PRF1 mutations [21]. Liu et al. proposed that the age of onset of familial HLH type 2 (FHL2) correlates with the degree of PRF1-dependent cytotoxicity impairment caused by specific PRF1 mutations [22]. While nonsense mutations predominate in pediatric HLH, missense mutations are more common in adult HLH, potentially contributing to delayed disease onset [21].

A parallel discussion concerns HLH cases following COVID-19 vaccination. By February 2023, 25 cases of HLH associated with COVID-19 vaccination were reported, with only one patient previously diagnosed with familial HLH type 3 [23]. The review revealed that both mRNA and viral vector vaccines were implicated, with 14 cases associated with Comirnaty (including 4 after the second dose), 5 with Vaxzevria (only after the first dose), and 3 with Moderna (COVID-19 mRNA-1273 vaccine). The median time to symptom onset was 16.5 days, and most cases received steroid therapy. Among these cases, three fatalities were reported, including one due to encephalopathy and shock.

Kim et al. described a case similar to ours, involving a 58-year-old patient who developed HLH following the first dose of the AstraZeneca (ChAdOx1) vaccine. The patient initially presented with neurological symptoms and cognitive deterioration, suspected to be ADEM, and was treated with methylprednisolone and intravenous immunoglobulins [24]. However, fever and pancytopenia emerged, leading to an HLH diagnosis. The patient ultimately died due to multiple organ failure (MOF), with no genetic testing conducted to confirm a hereditary form.

Another case reported by Yifan He [25] described a patient diagnosed with familial HLH following the first dose of Comirnaty. The patient developed annular erythema after the initial dose, which recurred post-boost, followed by fever and cytopenia. Further investigations confirmed familial HLH type 3 due to UNC13D gene mutation. The patient subsequently underwent an allogeneic transplant and remains in follow-up.

These cases highlight the potential role of COVID-19 vaccination in unmasking previously undiagnosed hereditary HLH. Although HLH remains a rare complication, its potential severity necessitates awareness.

Nevertheless, cases of neurological complications related to COVID-19 vaccination have been reported in the literature without the development of HLH, including cases of demyelinating disorders, acute disseminated encephalomyelitis (ADEM), other immune-mediated neurological syndromes, and even Susac syndrome [26,27].

Neurological presentations of HLH, while uncommon, lack pathognomonic cerebral imaging findings, further complicating early diagnosis. Enhanced recognition of HLH in the differential diagnosis of post-vaccine inflammatory syndromes is crucial to facilitate timely diagnosis and management.

## 4. Conclusions

The case report highlights an atypical presentation of a rare disease, representing the first known instance of primary HLH following COVID-19 vaccination with an initial neurological onset characterized by progressive gait ataxia, balance impairment, and cranial nerve involvement.

MRI findings demonstrated a highly pleomorphic pattern, with involvement of both white and gray matter, including hemorrhagic lesions, encephalic tumefactive lesions, and extensive longitudinal lesions of the spinal cord, with ring or nodular enhancement. No pathognomonic imaging features for adult CNS-HLH exist, and understanding of adult-onset primary HLH and the role of missense mutations remains limited. Some hypotheses suggest a “double hit” mechanism may trigger disease onset, as possibly observed in our patient.

Given the rarity of adult HLH, treatment follows pediatric protocols, and our patient achieved clinical and radiological disease control after undergoing an allogeneic transplant from an unrelated donor, despite the limited duration of the response.

This case emphasizes the importance of early recognition and aggressive management of HLH in adults presenting with neurological symptoms and contributes to the understanding of rare immune-mediated events potentially associated with vaccination.

Nevertheless, we acknowledge that a single case cannot establish a causal relationship between HLH and SARS-CoV-2 vaccination.

## Figures and Tables

**Figure 1 diagnostics-15-03000-f001:**
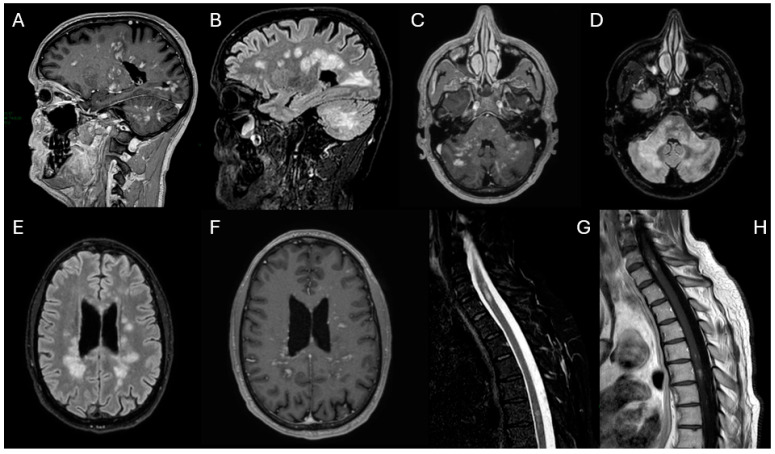
Contrast-enhanced T1-weighted (**A**,**C**,**F**,**H**), fluid-attenuated inversion recovery (FLAIR) (**B**,**D**,**E**) and STIR (**G**) images from the patient at baseline (October 2021). The images show multiple hyperintense FLAIR and contrast-enhancing white matter lesions with an asymmetric and periventricular distribution in the brain. Similar lesions are present in the spinal cord.

**Figure 2 diagnostics-15-03000-f002:**
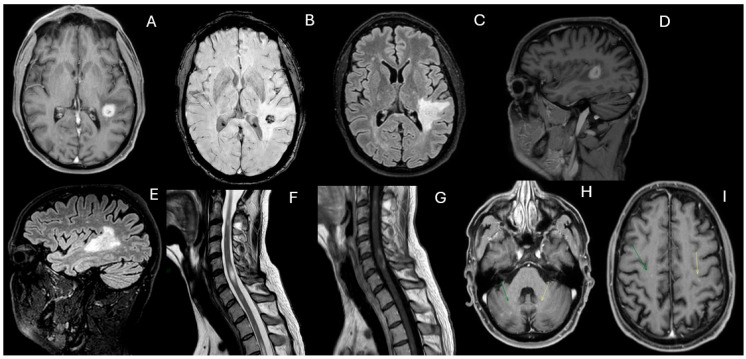
Contrast-enhanced T1-weighted (**A**,**D**,**G**–**I**), fluid-attenuated inversion recovery (FLAIR) (**C**,**E**), T2 (**F**) and SWI (**B**) images 10 months after. After clinical worsening following corticosteroid therapy, the images reveal the appearance of a lesion with “mass-like” features in the left paratrigonal region (**A**–**E**), with associated hemorrhagic components (**B**). Another similar lesion is present at the C6 level in the spinal cord (**F**,**G**). Several punctate contrast-enhancing lesions are still present in the cerebellum (**H**), in the supratentorial white matter (**I**) and in the spinal cord (**G**).

**Figure 3 diagnostics-15-03000-f003:**
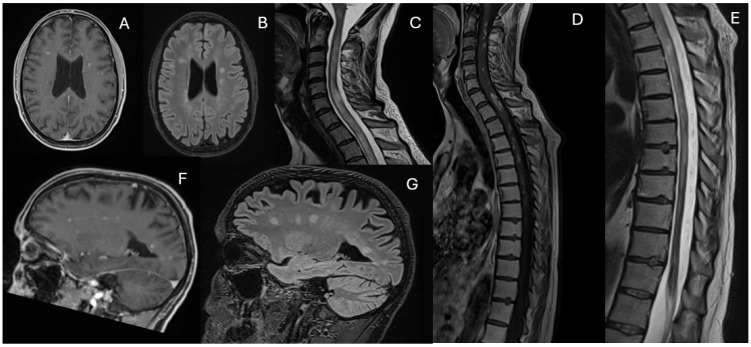
Contrast-enhanced T1-weighted (**A**,**D**,**F**), fluid-attenuated inversion recovery (FLAIR) (**B**,**G**) and T2 (**C**,**E**) images 9 months after. Following treatment with cyclophosphamide and mycophenolate mofetil, which resulted in clinical and radiological stability, in May 2023, the patient experienced a recurrence of a left-sided motor-sensory hemisyndrome. A new MRI was performed, revealing the reappearance of numerous hyperintense FLAIR (**B**,**G**) and contrast-enhancing (**A**,**F**) lesions in the deep white matter, along with extensive spinal cord involvement, showing a sub-continuous signal alteration from C2 to D8 (**C**,**E**) with multiple areas of contrast enhancement (**D**). Following this hospitalization, the diagnosis of hemophagocytic lymphohistiocytosis was made.

**Figure 4 diagnostics-15-03000-f004:**
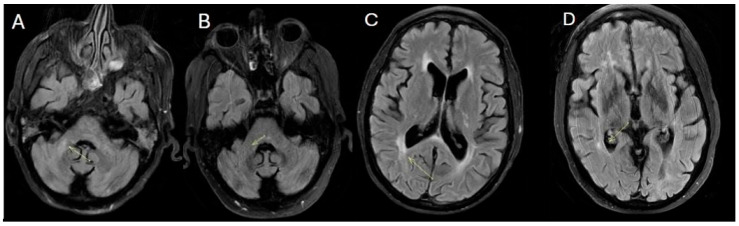
Fluid-attenuated inversion recovery (FLAIR) (**A**–**D**) (March 2025). The images show the appearance of a new hyperintense signal alteration on FLAIR in the right middle cerebellar peduncle and an increase in the hyperintense signal alteration on FLAIR in the right paratrigonal region.

## Data Availability

The data used and analyzed in this study are available from the corresponding author on reasonable request, due to privacy reason.

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
