# Peer review of "Adult Onset of Type 2 Familial Hemophagocytic Lymphohistiocytosis After SARS-CoV-2 Vaccination with an Unusual Neurological Onset: The Great Mimic"

_diagnostics, 2025, doi:10.3390/diagnostics15233000_

Round 1
Reviewer 1 Report
Comments and Suggestions for Authors
This article is a case report of a patient with hemophagocytic lymphohistiocytosis (HLH) who developed after receiving the SARS-CoV-2 vaccine. I believe the introduction is too brief and inadequate, lacking a systematic explanation of the pathogenesis, precipitating factors, and current research progress of this disease.
From a novel perspective, this article's innovation is limited; a single case alone cannot prove a causal relationship between HLH and the SARS-CoV-2 vaccine.
In addition, the formatting of some figures is not standardized. For example, Figures 1–3 contain unnecessary white space, which should be cropped or resized to improve the professional presentation.
Author Response
Thank you very much for taking the time to review this manuscript. Please find the detailed responses below and the corresponding revisions/corrections highlighted in the re-submitted files.
Comments 1: This article is a case report of a patient with hemophagocytic lymphohistiocytosis (HLH) who developed after receiving the SARS-CoV-2 vaccine. I believe the introduction is too brief and inadequate, lacking a systematic explanation of the pathogenesis, precipitating factors, and current research progress of this disease.
Response 1:
Thank you for pointing this out. We extended the paragraph on pathogenesis of HLH in the introduction on page 2 lines 50-86
Comments 2: From a novel perspective, this article's innovation is limited; a single case alone cannot prove a causal relationship between HLH and the SARS-CoV-2 vaccine
Response 2: We agree that a single case cannot establish a causal relationship between HLH and SARS-CoV-2 vaccination, however neurological disease onset following SARS-CoV-2 vaccination has been reported in the literature, including cases of demyelinating disorders, acute disseminated encephalomyelitis (ADEM), other immune-mediated neurological syndromes, and even Susac syndrome. These reports support the rationale for discussing a potential temporal association, although causality cannot be inferred. We have added a brief discussion of these reports in the revised manuscript, page 9, lines 298-301 and lines 324-325.
Comments 3: In addition, the formatting of some figures is not standardized. For example, Figures 1–3 contain unnecessary white space, which should be cropped or resized to improve the professional presentation.
Response 3: We edited the figures to improve their presentation.
Reviewer 2 Report
Comments and Suggestions for Authors
In their case report "Adult Onset of Type 2 Familial Hemophagocytic Lymphohistiocytosis After SARS-CoV-2 Vaccination with an unusual neurological onset: The Great Mimic," Bellisario et al. describe an exciting, albeit ultimately fatal, course of HLH induced by a COVID-19 vaccine in the presence of various mutations in the PRF1 gene.
The PRF1 gene variants were previously classified as class 3 mutations. Was immunophenotyping also performed for perforins in circulating blood? Was only the PRF1 gene examined, or were others examined as well, such as UNC13D, STX11, STXBP2, RAB27A, and LYST?
A tabular overview of the specified laboratory parameters over time would be helpful to ensure better readability. Are there values for LDH and sIL2-R?
Is EBV serology performed at the beginning or during the course of treatment? Can primary EBV infection be ruled out by PCR? If not available, I still recommend an intensive discussion as to why this was not performed.
The following case report, which should be cited, also refers to the presence of concomitant PRF1 gene mutations: Stadermann et al Int. J. Mol. Sci. 2024, 25(5), 2762.
In addition, the following article should also be mentioned in the discussion (Cetica et al, J. Allergy Clin. Immunol. 2016, 137, 188–196.), as it presumably deals with the largest Italian registry study.
Author Response
Thank you very much for taking the time to review this manuscript. Please find the detailed responses below and the corresponding revisions/corrections highlighted in the re-submitted files.
Comments 1: In their case report "Adult Onset of Type 2 Familial Hemophagocytic Lymphohistiocytosis After SARS-CoV-2 Vaccination with an unusual neurological onset: The Great Mimic," Bellisario et al. describe an exciting, albeit ultimately fatal, course of HLH induced by a COVID-19 vaccine in the presence of various mutations in the PRF1 gene.
The PRF1 gene variants were previously classified as class 3 mutations. Was immunophenotyping also performed for perforins in circulating blood? Was only the PRF1 gene examined, or were others examined as well, such as UNC13D, STX11, STXBP2, RAB27A, and LYST?
Response 1:
Thank you for pointing this out. All other HLH-related genes analyzed (UNC13D, STX11, STXBP2, RAB27A, LYST, SH2D1A and XIAP) were negative for pathogenic or likely pathogenic variants. We added this information on page 6, lines 191-193.
Comments 2: A tabular overview of the specified laboratory parameters over time would be helpful to ensure better readability. Are there values for LDH and sIL2-R?
Response 2: We included a table illustrating the temporal changes in laboratory parameters, particularly LDH and sIL-2R levels, which is provided in the supplementary material.
Comments 3: Is EBV serology performed at the beginning or during the course of treatment? Can primary EBV infection be ruled out by PCR? If not available, I still recommend an intensive discussion as to why this was not performed.
Response 3: We tested the presence of EBV-DNA in whole blood at onset and during follow up. We added the information in the manuscript (page 5, lines 169-170) and in the supplemental material.
Comments 4: The following case report, which should be cited, also refers to the presence of concomitant PRF1 gene mutations: Stadermann et al Int. J. Mol. Sci. 2024, 25(5), 2762.
Response 4: Thank you for pointing this out, We added this reference as ref (17).
Comments 5:In addition, the following article should also be mentioned in the discussion (Cetica et al, J. Allergy Clin. Immunol. 2016, 137, 188–196.), as it presumably deals with the largest Italian registry study.
Response 5: We thank the reviewer for indicating this article. We now discuss the findings of this report on page 8 lines 255-262.(ref. 19)
Reviewer 3 Report
Comments and Suggestions for Authors
• What is the main question addressed by the research?
1. Primary HLH is associated with mutations in one of nine known genes! Please explain more, add genes here.
2. Please add the neurological sypmtomes which you found to patients.
• Do you consider the topic original or relevant to the field? Does it address a specific gap in the field? Please also explain why this is/ is not the case.
From my opinion is OK. I didn’t detect any plagiarism.
• What specific improvements should the authors consider regarding the methodology?
I have made question which the authors need to give answer.
• Are the conclusions consistent with the evidence and arguments presentedand do they address the main question posed? Please also explain why this is/is not the case.
From my opinion yes.
• Are the references appropriate?
From my opinion yes.
• Any additional comments on the tables and figures.
No I do not have any other comments.
Author Response
Thank you very much for taking the time to review this manuscript. Please find the detailed responses below and the corresponding revisions/corrections highlighted in the re-submitted files.
Comments 1. Primary HLH is associated with mutations in one of nine known genes! Please explain more, add genes here.
Response 1: Thank you for pointing this out, we added the information on the mutational screening panel page 6 lines 191-192
Comments 2. Please add the neurological sypmtomes which you found to patients.
Response 2: We thank the reviewer for this comment. We have now included a detailed description of the neurological symptoms observed in the patient. These neurological findings have been integrated into the manuscript in the first two paragraphs of the case description, providing a clear account of the onset and evolution of the patient’s symptoms. We added the information on page 3 lines 89-97.
Round 2
Reviewer 1 Report
Comments and Suggestions for Authors
The authors have addressed all my questions. I have no further comments.
Reviewer 2 Report
Comments and Suggestions for Authors
All recommendations have been applied. I suggest to publish the manuscript in the present form. Congratulations to the nice case report.